# Trauma-Associated Pulmonary Laceration in Dogs—A Cross Sectional Study of 364 Dogs

**DOI:** 10.3390/vetsci7020041

**Published:** 2020-04-12

**Authors:** Giovanna Bertolini, Chiara Briola, Luca Angeloni, Arianna Costa, Paola Rocchi, Marco Caldin

**Affiliations:** 1Diagnostic and Interventional Radiology Division, San Marco Veterinary Clinic and Laboratory, via dell’Industria 3, 35030 Veggiano, Padova, Italy; chiara.briola@gmail.com (C.B.); luca.angeloni@sanmarcovet.it (L.A.); aryjorgo@yahoo.it (A.C.); 2Intensive Care Unit, San Marco Veterinary Clinic and Laboratory, via dell’Industria 3, 35030 Veggiano, Padova, Italy; rocchipmr@gmail.com; 3Clinical Pathology Division, San Marco Veterinary Clinic and Laboratory, via dell’Industria 3, 35030 Veggiano, Padova, Italy; mc@sanmarcovet.it

**Keywords:** thoracic trauma, blunt trauma, thoracic CT, cystic lesion, laceration running head, pulmonary laceration in dogs

## Abstract

In this study, we describe the computed tomography (CT) features of pulmonary laceration in a study population, which included 364 client-owned dogs that underwent CT examination for thoracic trauma, and compared the characteristics and outcomes of dogs with and without CT evidence of pulmonary laceration. Lung laceration occurred in 46/364 dogs with thoracic trauma (prevalence 12.6%). Dogs with lung laceration were significantly younger than dogs in the control group (median 42 months (interquartile range (IQR) 52.3) and 62 months (IQR 86.1), respectively; *p* = 0.02). Dogs with lung laceration were significantly heavier than dogs without laceration (median 20.8 kg (IQR 23.3) and median 8.7 kg (IQR 12.4 kg), respectively *p* < 0.0001). When comparing groups of dogs with thoracic trauma with and without lung laceration, the frequency of high-energy motor vehicle accident trauma was more elevated in dogs with lung laceration than in the control group. No significant differences were observed between groups regarding tge frequency and length of hospitalization and 30-day mortality. Similar to the human classification scheme, four CT patterns are described in dogs in this study: Type 1, large pulmonary laceration located deeply in the pulmonary parenchyma or around an interlobar fissure; Type 2, laceration occurring in the paraspinal lung parenchyma, not associated with vertebral fracture; Type 3, subpleural lung laceration intimately associated with an adjacent rib or vertebral fracture; Type 4, subpleural lesions not associated with rib fractures. Complications were seen in 2/46 dogs and included lung abscess and collapse.

## 1. Introduction

Injuries to the thorax are common in trauma patients and can affect any one or all components of the thoracic wall and thoracic cavity. Thoracic blunt trauma is more common and its causes include being hit by a car, falling from a height, and physical injuries from animal–animal or human–animal interactions [1,2,3]. Pulmonary contusion, implying traumatic injury to the alveoli and interstitium with alveolar hemorrhage and edema, but without significant alveolar disruption, is the most common injury of the lung parenchyma reported after blunt thoracic trauma in humans, as well as in small animals [4,5,6,7,8]. Pulmonary laceration is a tear of the lung parenchyma due to higher-energy blunt trauma than contusion and leads to a mechanical shear or puncture, disrupting the lung parenchyma. Because of the normal pulmonary elastic recoil, the tissues surrounding the cleavage plane pull back from it, resulting in round or oval space lesions on imaging, instead of the linear appearance typically seen in other solid organs [9,10,11]. Other names that have been used in the radiology literature include traumatic bullae, traumatic pneumatoceles, haematocele or traumatic lung cyst or pseudocysts [12,13]. In this manuscript, the authors will use the term pulmonary laceration because it unambiguously describes the nature of the lesion in the lung parenchyma and allows a consistent, interdisciplinary comparison with the fundamental radiology literature [5,9]. In humans, computed tomography (CT) is the recommended imaging method for pulmonary laceration detection and characterization [5]. Recently, it has been documented that thoracic radiography has a lower sensitivity than CT in detecting lesions related to blunt thoracic trauma in veterinary patients [14]. In the veterinary literature, CT features of pulmonary laceration in dogs have been described in a book and recently reported in single case reports [15,16,17]. Different types of lung laceration have been described using CT in people, according to the mechanism of injury (compression rupture, compression shear, rib penetration tear, or adhesion tear), CT pattern, and location within the lung [5]. To the authors’ knowledge, similar descriptions have not been reported in dogs thus far.

The aim of this study is to describe the CT features of pulmonary laceration in a population of traumatized dogs and compare the characteristics and outcomes between dogs with and without CT evidence of pulmonary laceration.

## 2. Material and Methods

### 2.1. Study Design, Sample and Setting

This is an observational, cross-sectional study. The study population included client-owned dogs that underwent multidetector-row computed tomography (MDCT) examination for trauma. All collection procedures were performed solely for the dog’s benefit and for standard diagnostic and monitoring purposes. Previous informed written consent was obtained from all dog owners. All the procedures performed complied with the European legislation “on the protection of animals used for scientific purposes” (Directive 2010/63/EU) and with the ethical requirement of the Italian law (Decreto Legislativo 04 March 2014, n. 26). All dogs with trauma of any origin (motor vehicle accident, falling from a height, and thoracic bite trauma) that had undergone CT examination at our clinic from 1st January 2005 until 30th August 2019 at the Diagnostic and Interventional Radiology Division of the San Marco Veterinary Clinic were considered for inclusion in the study. Eligibility criteria and information required were: (1) MDCT scans of the body including the whole thorax, (2) complete anamnestic, clinical and clinic–pathological evaluation (for further study; results will not be reported here), (3) frequency and length of hospitalization and (4) 30-day mortality. Information on medical therapy, surgical or interventional procedures, and follow-up were all available for the study. Radiographic studies were also evaluated when available. The study group included dogs that underwent CT examination for trauma with a CT diagnosis of pulmonary laceration. The control group included dogs that underwent CT examination in the same timeframe for thoracic trauma without evidence of pulmonary laceration.

Only the incident cases (newly diagnosed) were included in the study group. Hence, for dogs with repeat imaging after an initial MDCT diagnosis of trauma, only the first examination was recruited for image and statistical analyses. Follow-up imaging was analyzed when available to assess changes in the pulmonary lesions over time.

### 2.2. MDCT Scanning Techniques

MDCT examinations were performed either using a 16-MDCT scanner (GE Medical Systems, Lightspeed 16, Milan, Italy) from 2005 to 2013 or a Dual Source CT scanner from 2014 to 2019 (128 × 2 DSCT, Somatom Definition Flash or 192 × 2 Somatom Force, Siemens, Erlangen, Germany). All patients were awake and were positioned in lateral or sternal recumbency on the CT table. The scan parameters for 16-MDCT were as follows: acquisition parameters, helical modality, detector configuration 16 × 1.25 mm, pitch 0.938:1 and 0.7 s rotation time. Dose parameters were 120 kV and 160–230 mAs depending on the dog’s size. Images were reconstructed using a soft-tissue reconstruction algorithm with a 512^2^-matrix size.

DSCT examinations were performed with the following settings: one tube, 120 kVp, 400 mAs/rot (0.28 s), collimation 128/192 × 0.6 mm. Alternatively, a high-pitch Flash modality scan protocol was used, with two tubes set at the same energy level (120 kVp), a gantry rotation time of 0.28 s/r, a pitch of 3.4 and 2 × 128/192 slices. The reconstructed slice thickness was 0.6 mm, with a 0.3 mm interval, using both protocols.

### 2.3. Image Analyses

In all affected dogs, the characteristics of the lung laceration were described at the time the CT report was done and assessed on the basis of the index case suspected on CT images in 2005, then confirmed by the surgical and histopathological results.

At our center, all of the information collected from different divisions on an individual patient is registered in a database and classified using selected keywords. The software (version 153, POA System 9.0, Copyright © 1993–2016) allows a search for data that meet specific criteria. The CT reports’ database runs on the same database and keywords are entered into the system each time by radiologists when reporting. In this case, the keywords inserted in the software and used to recruit the population were ‘thoracic trauma’ (all dogs underwent CT examination for thoracic trauma in the same timeframe) and/or ‘lung laceration’ (study group). For further characterization of the lung lesions, CT examinations were all retrieved from the Picture Archiving Communication System (PACS) and analyzed by one author (GB) using a dedicated freestanding workstation and vendor-specific postprocessing software (Syngo.Via, Siemens, Germany). The original data set and a combination of two-dimensional (2D) multiplanar reformations (MPRs), and three-dimensional (3D) volume-rendered (VR) postprocessing techniques were used as part of a routine assessment of MDCT volume data.

Pulmonary lacerations were categorized into four types similar to the human classification scheme proposed by Wagner et al. [5]: Type 1, large pulmonary laceration located deeply in the pulmonary parenchyma or around an interlobar fissure; Type 2, laceration occurring in the paraspinal lung parenchyma, not associated with vertebral fracture; Type 3, subpleural lung laceration intimately associated with an adjacent rib or vertebral fracture; Type 4, subpleural lesions not associated with rib fractures. Concomitant thoracic lesions were recorded as well if the lacerations were single or multiple, with features similar to the human classification.

### 2.4. Statistical Methods

A univariate analysis between dogs with lung laceration and the control group was performed using a chi-square test for the variables gender, sexual status in males and females, hospitalization frequency, and 30-day mortality.

Mann–Whitney U test was used to analyze the difference for age, body weight, temperature, pulse rate, respiratory rate, and length of hospitalization between the two groups of dogs. The *t* test was used to compare the mean blood pressure of dogs.

Furthermore, backward stepwise multivariable logistic regression analysis was carried out to identify factors associated to lung laceration (age, body weight, gender, sexual status in males and females, temperature, pulse rate, respiratory rate, mean blood pressure, frequency and length of hospitalization, and 30-day mortality). For all statistical analyses, the significance level was set to α = 0.05.

## 3. Results

### 3.1. Characteristics of the MDCT Study Population

In the selected period of time, 364 dogs that underwent CT for trauma at our center matched the inclusion criteria: 220 (60.4%) were males (196 intact and 24 castrated) and 144 (39.6%) were females (89 intact and 55 neutered). Represented breeds are reported in Table 1. The frequencies of different types of trauma in the study group and control population are shown in Scheme 1.

Of the 364 dogs with trauma, 46 showed CT signs of lung laceration (prevalence 12.6%). There was no statistical difference regarding the gender (*p* = 0.69) and sexual status between the 46 dogs with lung laceration and the remaining 318 dogs without laceration (*p* = 0.91 and *p* = 0.42 in males and females, respectively). The median age of the dogs with laceration was 42 months (interquartile range (IQR) 52.3 months) and that of traumatized dogs without laceration was 62 months (IQR 86.1 months). This difference was statistically significant (*p* = 0.02). (Figure 1). Dogs with lung laceration were significantly heavier than dogs without laceration (median 20.8 kg (IQR 23.3) and median 8.7 kg (IQR 12.4 kg), respectively; *p* < 0.0001) (Figure 2). The backward stepwise multivariable logistic regression analysis identified only age and body weight as factors associated with lung laceration (0.01 and <0.0001, respectively).

The frequency of hospitalization was 84.8% in the group of dogs with pulmonary laceration (39/46) and 82.4% (262/318 dogs) in the control group. This difference was not statistically significant (*p* = 0.68). No differences were observed between groups regarding the length of hospitalization (median 5 days for both groups; *p* = 0.45). Table 2 shows the mean and median values of all tested factors in two groups.

The 30-day mortality was 6/46 in dogs with laceration (13%) and 48/318 (15.1%) in control dogs. This difference was not statistically significant (*p* = 0.71). The multivariable logistic regression analysis showed that age, body weight, gender, sexual status in males and females, temperature, pulse rate, respiratory rate, mean blood pressure, frequency and length of hospitalization were not risk factors predicting death.

### 3.2. Characteristics of Lung Laceration

Initial CT examinations in dogs affected by lung laceration were obtained between 1 h and 10 days after the traumatic injuries. In 24 of the 46 dogs, CT images were obtained within 24 h of trauma. Twenty-two of the 46 dogs with pulmonary lacerations also had a thoracic x-ray examination available. Lesions due to pulmonary lacerations were detected via radiography in 13/22 dogs. In the remaining patients, cavitary lesions were not observed on admission radiographs. At CT, 13/46 dogs had Type 1 lacerations located deep in the pulmonary tissue or near to the interlobar fissures. Five/46 dogs had subpleural lung lacerations not associated with vertebral fractures (Type 2) (Figure 3 and Figure 4). Eleven patients had Type 3 lung lacerations affecting the subpleural lung parenchyma in correspondence to the rib fractures (Figure 5A). Four dogs showed subpleural lung laceration not associated with rib fracture (Figure 5B). Finally, 14/46 dogs had a mixed pattern including more than one type of lung laceration. Lung lacerations of Type 1 were always ovoid in shape, with regular margins. They contained air or showed hydro-fluid levels of blood and air in different proportions. Other types of lacerations showed rounded, irregular or slit-like shapes and were smaller than Type 1 lacerations. CT signs of pulmonary contusion always surrounded the lung lacerations, and alveolar hemorrhage in most cases. Pneumothorax always accompanied Type 3 lacerations and irregularly accompanied other types of lacerations (Figure 6).

One or more follow-up CT studies were available in four cases from 2 days to one month after the initial CT examination. In follow-up CT studies, lesions showed increasing fluid content over time (Figure 7). Complications of lung lacerations were observed in two cases. One dog showing several lung lacerations in the right caudal lobe one hour after trauma underwent lobectomy few days after admission for the abscessation and pneumonic involvement of that lobe. Another dog showed a lung collapse due to massive pneumothorax for a ruptured subpleural lung laceration (Figure 8). In this case, a chest drainage system placed for few days allowed the lung to be pulled up against the parietal pleura and the dog was discharged to its home in a stable condition.

## 4. Discussion

The prevalence of lung laceration in this study was higher than that reported previously in the veterinary literature with conventional radiology (12.6% vs. 10%) [2]. In humans, CT is reported to have a greater sensitivity to the diagnosis of lung laceration than radiography (100% and 20% respectively) [11]. It has been reported that only 50% of pulmonary lacerations can be detected on radiographs within 24 h from the traumatic event, mainly because the surrounding pulmonary contusion can mask the cavitary lesion. Similar studies are not reported in small animals. However, a recent study compared thoracic radiology and CT diagnostic performances in blunt trauma caused by motor vehicle accidents in canine patients for detecting lung contusions, pneumothorax, pleural effusion, and rib fractures in dogs [14]. Results showed that radiology is less sensitive than CT at detecting these injuries. In the present study, 22 dogs had also thoracic X-ray available and only 59% (13/22) showed one or more cavitary lesions, consistent with lung laceration. As has been reported in the human radiology literature [18], over time, the disrupted pulmonary tissue filled with blood and/or air and was visible on thoracic imaging as one or more pulmonary parenchymal cavities (Figure 9), appearing as gas–fluid levels, with surrounding pulmonary consolidation and ground-glass opacity related to hemorrhage and atelectasis. The time of the initial CT scans varied between 1 h and 10 days after the traumatic injuries. This wide temporal range might have affected the results (e.g., lesions that were not yet visible or had already disappeared).

Although the exact mechanism leading to these cavitary lesions is still unknown, most authors agree that compressive traumatic forces would increase intrapulmonary pressure until the tearing of the lung parenchyma. Sudden decompression, increasing negative intrathoracic pressure, would allow the elastic lung tissue to recoil, leading to the formation of cavitary lesions. The median age of dogs with pulmonary laceration in this study was significantly lower than that of traumatized dogs without laceration (47 vs. 62 months respectively; *p* = 0.02). This could reflect, as reported in humans, the greater flexibility of the chest wall of younger subjects, resulting in a higher likelihood of lung injury with high-energy trauma [19,20,21]. From our results, dogs with lung laceration are significantly heavier than dogs without laceration (median of dogs with lung laceration 20.8 kg versus median 8.7 kg of traumatized dog without lung laceration; *p* < 0.0001). The morphology type of the thorax might play a role in the laceration of the lung parenchyma after blunt trauma, but the sample size per breed was too small to formally assess this.

Different types of lung lacerations, probably reflecting different pathogenetic mechanisms, have been recognized here, often coexisting in the same patient or lobe [5,22,23,24,25,26]. Type 1 and Type 3 were found more frequently in the population of this study. Shear between parenchyma and tracheobronchial structures may be the mechanism leading to deep parenchymal Type 1 lacerations [5]. In our patient cases, Type 1 lesions were always accompanied by parenchyma contusion. Pneumothorax and/or pneumomediastinum were seldom present in these Type 1 cases. Type 2 lung lacerations were paravertebral subpleural lesions not associated to vertebral or costal fracture, but probably resulting from lateral compression forces between the lung and the thoracic spine [22]. In one case, Type 2 was associated to vertebral subluxation. Type 3 lung lacerations were strictly associated to rib fracture and accompanied by lung contusion/hemorrhage and pneumothorax. Type 4 lesions have been classified those pulmonary subpleural cavitary lesions not associated with rib fracture. In humans, these lesions are thought to form in sites of previous pleuro-pulmonary adhesion [5,26,27,28]. Here, a mechanism could not be hypothesized. Many dogs included in this study had concomitant other thoracic or extra-thoracic lesions and required interventions that might have influenced the outcome. This represents a limitation of the present work.

In people, pulmonary lacerations often resolve spontaneously over a few weeks, although they may be rarely complicated by a pulmonary abscess or a bronchopleural fistula [24,29]. Serious complications are reported in people when blood that leaks from the injured lung parenchyma and peripheral pulmonary circulation enters the airway or thoracic cavity leading to hemoptysis or hemothorax. In some cases, these patients may require lung resection to remove severely injured lung tissue, to control hemorrhage, or to remove irreparable proximal bronchus injuries [18,30]. A lobectomy was necessary only in one dog in the study group due to abscessation and pneumonic involvement of the lacerated lung lobe.

In conclusion, lung laceration can frequently occur in traumatized dogs—large breeds and young patients in particular. In this study, we described four different types of lacerations as they appear on CT imaging.

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
