# Peer review of "Trauma-Associated Pulmonary Laceration in Dogs—A Cross Sectional Study of 364 Dogs"

_vetsci, 2020, doi:10.3390/vetsci7020041_

Round 1
Reviewer 1 Report
Thank you for preparing this manuscript describing CT findings of patients with trauma and how the lung pathology changes coincide with a human description of pulmonary lacerations.
I have some major concerns about the study design and how the conclusions were reached. The materials and methods aren't quite clear on how the cases were chosen and very little information is given as to what interventions were performed. Unfortunately, there are then very strong statements in discussion and conclusion about how these cases are diagnosed and managed, which is inappropriate given the information provided.
If this is meant to be an imaging description study, then it can be valuable. However, when the authors move into the realm of recommending treatments and prognosis associated with them, it is an overstep given what is provided in the manuscript.
If the authors can lessen the conclusions and frame this as a purely descriptive study on CT findings in dogs with SUSPECTED pulmonary lacerations it may be publishable. Please find more targeted comments below.
Lines 39-41: this sentence is confusing; is than supposed to be then? As in , contusion happens subsequent to laceration?
Line 41-42: is cleavage plane the same as the area of lung laceration. Since this is a new term please modify.
Lines 43-45: It sounds like here you are saying that the veterinary literate incorrectly defines these lesions as bullae, pneumatoceles, hematocele, cyst or pseudocyst. It may be best to transition the previous statement into one that suggests all of these lesions (laceration and the other veterinary terms) are described in the same way, yet pulmonary laceration is the most correct term. Is there human literature that confirms that imaging findings are actual lacerations during surgery? One of your references here is a human article detailing pseudocysts.
Lines 61-82: It is unclear here, is this a retrospective study? Did you have a protocol in place to perform these diagnostics since 2005 and hence it is prospective? Please clarify what the basic study design is.
Lines 75-78: how were the lung lacerations confirmed? Was any intervention included in inclusion criteria? All blunt trauma? Penetrating trauma included? Need more information please.
Lines 79: What does "incident cases" mean? Is this referring to the CTs performed following the traumatic event? Please clarify.
Lines 98-99: What does this line mean? All cases were compared to a single case from 2005? Its not clear what is meant by this statement.
Lines 99-107: IS this sentence stating that all cases were retrospectively searched in PACS using the term lung laceration? Additionally were all found cases then re-reviewed by a single person (the person stated in the paper?) This entire paragraph is hard to follow, please clarify for ease of understanding.
Lines 98-114: How were the control group cases defined and enrolled? Here you only describe how you enrolled the lung laceration group, however almost all of your cases in this study are control cases without lung laceration. How were those injuries classified?
Lines 123-124: What pressure? Blood pressure? Ventilatory support pressure? Please clarify.
Lines 125-126: logistic regression was performed on what factors? Those listed above which are predominantly signalment and basic examination findings? Where factors such as type of trauma, time from trauma to admission, etc considered. Please list though factors used in analysis here.
Lines 133-144: Did any dogs undergo intervention? Why isn't the type of trauma listed and included in analysis? This would seem to be a major concern and listing/including this in analysis would be helpful to readers.
Lines 156-160: Were any interventions related to mortality? This seems like a major omission.
Line 168: Lesions detected via radiography?
Lines 165-167: what was the median or mean time from trauma to CT? did you notice a difference in appearance of lung laceration depending on time which the CT was performed?
169-175: please include % here instead of just numbers.
Figure 3: Please alter photos to show/point out the deeper lesions and the areas of fluid/blood. In addition you need to add left and right markers to assure the readers are viewing it the same way. Please list side markers for ALL images which show both sides.
Figure 4: is the laceration in the lung lobe near the luxation?
Figure 7: was there any intervention in this case? (chest tube, surgery, etc)
Lines 219-221: What was the percentage in your study of radiographically identified lesions?
Lines 222-224: Is there a citation for this comment? Human or veterinary?
Lines 248-250: You didn't report breed type in this study, if chest wall conformation is a contributor, can that information be added then instead of speculation you have some information to back that claim.
Line 256: Does "these cases" refer to those with type 1 lesion? Please clarify.
Lines 262-263: why is it hypothesized, because humans suspect that method? Isn’t it more appropriate to suggest no mechanism can be hypothesized as similar to humans?
Line 267: ender the thoracic what? Is there are word missing in this sentence? Perhaps thoracic cavity?
Line 270: Abscessualization? Would suggest abscessation or abscess formation in a lung lobe.
Line 272: Careful with the comment on large breed dogs here, you have no breeds listed. I've seen 20kg English Bulldogs. Please list breeds if you are making these conclusions.
Lines 272-273: Also, you note only blunt trauma as the source here, please assure that the information on it only being blunt trauma is included throughout the manuscript. How do you know it was high-energy? You have no information provided on what type of trauma the patients sustained. Either include this information to substantiate your claims, or remove this from conclusion.
Lines 274-275: Misinterpretation of what? You are strictly providing an imaging description. You have provided no proof with necropsy, histopathology, or surgical exploratory that the lungs were truly lacerated. What do you suggest they are differentiated from? Ruptured bullae? You can't claim that with the information in this study. You should lessen your conclusions here to show that you have only provided imaging descriptions for SUSPECTED lacerations as you have given no definitive information.
Lines 276-277: This comment is also less-than appropriate given the information provided. How many of these cases had chest wall defects, flail chest, pseudoflail chest, more than one rib fracture? You have given the reader NO information to support a claim that conservative management is best for these cases. This comment needs to be removed unless the authors can support this claim with clinical data.
Author Response
Authors’ reply to the Reviewer1
First of all, Authors would like to thank you the Reviewer for the time spent for the revision and comments and advice. Authors agree with most of them and changed accordingly the manuscript.
In particular, as this is an imaging description study, Authors removed any recommendation from the discussion and conclusion, in the revised manuscript.
Lines 39-41: this sentence is confusing; is than supposed to be then? As in, contusion happens subsequent to laceration? Thank you for the suggestion. According to the reviewer’s suggestion, we rephrased the sentences in the revised manuscript.
Line 41-42: is cleavage plane the same as the area of lung laceration. Since this is a new term please modify.
Thank you for this observation. Cleavage plane is not a new term, in truth. It is usually used to indicate the plane of separation between cells or tissues and is commonly used in anatomy, pathology, radiology (as in this case). Authors believe that readers of this Journal can understand this term.
Lines 43-45: It sounds like here you are saying that the veterinary literate incorrectly defines these lesions as bullae, pneumatoceles, hematocele, cyst or pseudocyst. It may be best to transition the previous statement into one that suggests all of these lesions (laceration and the other veterinary terms) are described in the same way, yet pulmonary laceration is the most correct term…
Authors did not mean that the veterinary literature is not correct. Authors chose the term ‘lung laceration’ and used the same term throughout the paper for consistency.
Is there human literature that confirms that imaging findings are actual lacerations during surgery?
Lung laceration is well described entity in human radiology literature and there are several papers reporting surgical (open surgery or thoracosopic) confirmation. In any case, there are two previous published reports in dogs (quoted in this article) describing the same features in dogs using CT. In addition, our first case (index case 2005) underwent surgical lobectomy (and histology). We reported more clearly this information in the revised text (where describe the index case), thank you.
Lines 61-82: It is unclear here, is this a retrospective study? Did you have a protocol in place to perform these diagnostics since 2005 and hence it is prospective? Please clarify what the basic study design is.
Thank you to the reviewer for this comment. In truth, the study design was declared at the end of the introduction (Line 57) and was defined following the STROBE-vet guidelines. However, we move this information into the proper section.
Lines 75-78: how were the lung lacerations confirmed? Was any intervention included in inclusion criteria? All blunt trauma? Penetrating trauma included? Need more information please.
Thank you to the reviewer, for these comments. At line 67-67 of Material and Methods said ‘All dogs with trauma of any origin that had undergone CT examination at our clinic from …’ However, we specify further in this new version of the paper.
We added more information regarding the types of trauma included (non-penetrating and penetrating). Interventions were not among the inclusion criteria.
Lines 79: What does "incident cases" mean? Is this referring to the CTs performed following the traumatic event? Please clarify
Incident is an epidemiological term. Incident cases comprise cases newly diagnosed during a defined time period (those that generate the incidence of a disease/condition). We added 'newly diagnosed' in parenthesis for clarity.
Lines 98-99: What does this line mean? All cases were compared to a single case from 2005? Its not clear what is meant by this statement.
The ‘index case’ refers to the first documented case. We describe better this in the new version. Thank you for your comment.
Lines 99-107: IS this sentence stating that all cases were retrospectively searched in PACS using the term lung laceration? Additionally, were all found cases then re-reviewed by a single person (the person stated in the paper?) This entire paragraph is hard to follow, please clarify for ease of understanding. Lines 98-114: How were the control group cases defined and enrolled? Here you only describe how you enrolled the lung laceration group, however almost all of your cases in this study are control cases without lung laceration. How were those injuries classified?
Thank you for these observations. Authors rephrase these sentences, providing also information for controls enrolment by keyword thoracic trauma.
Lines 123-124: What pressure? Blood pressure? Ventilatory support pressure? Please clarify.
Thank you, clarified.
Lines 125-126: logistic regression was performed on what factors? Those listed above which are predominantly signalment and basic examination findings? Where factors such as type of trauma, time from trauma to admission, etc considered. Please list though factors used in analysis here
Thank you to the reviewer for this observation. It was a mistake. Authors rewritten this period.
Lines 133-144: Did any dogs undergo intervention? Why isn't the type of trauma listed and included in analysis? This would seem to be a major concern and listing/including this in analysis would be helpful to readers.
Thank you to the Reviewer for this observation. In the revised version of the paper we include types of trauma studied and provided this information in material and methods, results, and discussion. We added a graph to show the frequencies or different types of trauma in the two populations.
Lines 156-160: Were any interventions related to mortality? This seems like a major omission.
This was not a subject of this paper. However, we agree with the Reviewer that results might be influenced by the treatment. We add this comment in the Discussion section.
Line 168: Lesions detected via radiography?
Clarified, thank you.
Lines 165-167: what was the median or mean time from trauma to CT? did you notice a difference in appearance of lung laceration depending on time which the CT was performed?
We provided information regarding the time for CT after trauma in the 3.2 Section of results. Half of dogs had first CT withing 24 hours from trauma. We subjectively noted a difference and reported this in Discussion see… ‘Over time, the disrupted pulmonary tissue fills with blood and/or air and manifests on thoracic imaging as one or more pulmonary parenchymal cavities (Figure 9)…’
Figure 3: Please alter photos to show/point out the deeper lesions and the areas of fluid/blood. In addition you need to add left and right markers to assure the readers are viewing it the same way. Please list side markers for ALL images which show both sides.
Figure 3: Please alter photos to show/point out the deeper lesions and the areas of fluid/blood. In addition you need to add left and right markers to assure the readers are viewing it the same way. Please list side markers for ALL images which show both sides. Made.
Figure 4: is the laceration in the lung lobe near the luxation? Indicated the vertebra level in A (transverse view).
Lines 219-221: What was the percentage in your study of radiographically identified lesions?
We add this information, thank you.
Lines 222-224: Is there a citation for this comment? Human or veterinary?
We specified in the revised paper and move the reference in the appropriate place.
Lines 248-250: You didn't report breed type in this study, if chest wall conformation is a contributor, can that information be added then instead of speculation you have some information to back that claim.
Line 256: Does "these cases" refer to those with type 1 lesion? Please clarify. Clarified.
Lines 262-263: why is it hypothesized, because humans suspect that method? Isn’t it more appropriate to suggest no mechanism can be hypothesized as similar to humans?
Modified the sentence as suggested.
Line 267: ender the thoracic what? Is there are word missing in this sentence? Perhaps thoracic cavity? We add ‘cavity’ thank you.
Line 270: Abscessualization? Would suggest abscessation or abscess formation in a lung lobe. Changed following the Reviewer’s suggestion. Thank you.
Line 272: Careful with the comment on large breed dogs here, you have no breeds listed. I've seen 20kg English Bulldogs. Please list breeds if you are making these conclusions.
We agree with the Reviewer. We add the list of breeds. Thank you.
Lines 272-273: Also, you note only blunt trauma as the source here, please assure that the information on it only being blunt trauma is included throughout the manuscript. How do you know it was high-energy? You have no information provided on what type of trauma the patients sustained. Either include this information to substantiate your claims, or remove this from conclusion.
We agree with the Reviewer. We added information regarding the types of trauma. Thank you.
Lines 274-275: Misinterpretation of what? You are strictly providing an imaging description. You have provided no proof with necropsy, histopathology, or surgical exploratory that the lungs were truly lacerated. What do you suggest they are differentiated from? Ruptured bullae? You can't claim that with the information in this study. You should lessen your conclusions here to show that you have only provided imaging descriptions for SUSPECTED lacerations as you have given no definitive information.
Lines 276-277: This comment is also less-than appropriate given the information provided. How many of these cases had chest wall defects, flail chest, pseudoflail chest, more than one rib fracture? You have given the reader NO information to support a claim that conservative management is best for these cases. This comment needs to be removed unless the authors can support this claim with clinical data.
Authors removed any comment, thought to be inappropriate by the Reviewer.
Reviewer 2 Report
I thank the authors for the study. Is is important in its field, and the manuscript is clear and easy to read with very good images. The study will help veterinary radiologists and clinicians to recognise lung lacerations and helps to bring terminology into line. I have some comments and suggestions listed below.
The Figures 1 and 2 are less informative and could be deleted.
L13: that underwent
L16: Change to “group without lung laceration”. Mean or median?
L46: because it (missing word?).
L50: lower
L86: I assume that all were awake? Maybe it could be said more clearly. All dogs (or patients?) were awake and were positioned…
Ls109-113: I would love to have references to images already here. It would help the reader to familiarize to the laceration types.
L129: P=
L133: that underwent
L136: no statistical
L165: The range between the CT scans was large (1h-10d). Could this have affected the results? Is it possible that in some cases the lesions were not yet visible, and in some they might have already disappeared? Add in discussion/limitations?
Ls 169-175: A table of these would be informative.
L220: The difference between the numbers is small, and probably insignificant. The ref 6 is not a veterinary article.
L273: Add that the types used were similar to ones used in human literature.
L341: Ref 26, the year is missing.
Table 1: Nonparametric and parametric have changed places. Delete “X2-square test”.
Figure 4: Was the level of the laceration also T12-13? Please add.
Author Response
Authors’ reply to the Reviewer2
The Authors would like to thank the Reviewer for the comments on this work.
The Figures 1 and 2 are less informative and could be deleted. Thank you for this comment. Authors rely on the Editor opinion for this decision.
L13: that underwent – Authors will add ‘that’
L16: Change to “group without lung laceration”. Mean or median? We add ‘lung’ and ‘median’, thank you.
L46: because it (missing word?). ‘it’ added, thank you.
L50: lower . Mistake, thank you.
L86: I assume that all were awake? Maybe it could be said more clearly. All dogs (or patients?) were awake and were positioned… We changed the sentence, thank you,
Ls109-113: I would love to have references to images already here. It would help the reader to familiarize to the laceration types. Authors agree with the Reviewer. However, images are results. We will rely on the Editor decision. Thank you.
L129: P= . Alfa is tested and P is the result.
L133: that underwent . Added that.
L136: no statistical – Corrected.Thank you.
L165: The range between the CT scans was large (1h-10d). Could this have affected the results? Is it possible that in some cases the lesions were not yet visible, and in some they might have already disappeared? Add in discussion/limitations? We add some observation on this in the discussion. Thank you
Ls 169-175: A table of these would be informative.
L220: The difference between the numbers is small, and probably insignificant. The ref 6 is not a veterinary article. Mistake, thank you.
L273: Add that the types used were similar to ones used in human literature. Made
L341: Ref 26, the year is missing. Mistake, thank you
Table 1: Nonparametric and parametric have changed places. Delete “X2-square test”. thank you
Figure 4: Was the level of the laceration also T12-13? Please add. – Added, Thank you.